# Behavioral differences at scent stations between two exploited species of desert canids

**Maksim Sergeyev**[ORCID]*, **Kelsey A. Richards, Kristen S. Ellis**[ORCID]**, Lucas K. Hall, Jason A. Wood**[ORCID]**, Randy T. Larsen**

Department of Plant and Wildlife Sciences, Brigham Young University, Provo, UT, United States of America

* ecomaksimsergeyev@gmail.com

## Abstract

Coyotes (*Canis latrans*) and kit foxes (*Vulpes macrotis*) are desert canids that share ecological similarities, but have disparate histories with anthropogenic pressure that may influence their responses towards novel stimuli. We used remote cameras to investigate response to novel stimuli for these two species. We predicted that coyotes (heavily pressured species) would be more wary towards novel stimuli on unprotected land (canid harvest activities are permitted) than in protected areas (canid harvest activities are not permitted), whereas kit foxes (less pressured species) would exhibit no difference. We examined differences in the investigative behaviors at 660 scent stations in both protected and unprotected areas. Coyotes showed no differences between protected and unprotected land and were generally more wary than kit foxes, supporting our prediction. Kit foxes were more investigative on protected land, contrary to our expectations. Our study provides evidence that anthropogenic pressure can alter the behaviors of wildlife species.

## Introduction

Behavioral responses of wildlife to novel anthropogenic objects vary greatly and can be influenced by social status, trophic level, past experiences with anthropogenic stimuli, and differences in personality [1–4]. Responses to novel objects are generally categorized as either neophilic or neophobic. Neophilia (attraction to novelty) can be an advantageous behavior in discovering new resources, related to the concept of boldness (tendency to take risks), however, increased conflict with humans may arise as animals interact with anthropogenic stimuli [4, 5]. Conversely, neophobia (fear of novel stimuli) is typically classified as gustatory (novel food sources), social (novel interactions between conspecifics), or predator [novel objects perceived as predatory threats; 2] Neophobia has been associated with lower trophic levels and social status, and can be influenced by familiarity with surroundings [1, 2, 6]. Repeated exposure to anthropogenic stimuli may cause habituation (decreased sensitivity to novel objects) or sensitization [increased avoidance; 4]. Consequently, prior interactions with anthropogenic disturbances can influence behavioral responses to novel stimuli.

**Data Availability Statement:** https://github.com/MaksimSergeyev/CanidBehaviorScentStations

**Funding:** We are very grateful to the U.S. Department of Defense, the Bureau of Land Management, the Utah Division of Wildlife

Resources, and Brigham Young University for their funding and support on this project.

**Competing interests:** The authors have declared that no competing interests exist.

Species subjected to intense anthropogenic pressure (e.g., hunting, trapping) may exhibit increased wariness than less pressured species [7–9]. If behaviors that render individuals susceptible to hunting and trapping by humans (e.g., investigating anthropogenic stimuli) have a genetic basis, these behaviors would be subjected to selection [10]. Thus, pressure towards hunted and trapped species could reduce the genetic availability of specific behaviors (that increase mortality) and, over generations, influence interactions with novel anthropogenic stimuli [9]. As a result, species with a history of anthropogenic pressure may exhibit increased neophobia.

Coyotes (*Canis latrans*) and kit foxes (*Vulpes macrotis*) are two canid species found across arid environments of North America [11, 12] that have ecological similarities but disparate histories of anthropogenic pressure that may influence their behaviors [13]. Coyotes, long considered a nuisance species, have been subjected to intense lethal control [14–19], potentially causing heightened neophobia [6, 20–29]. Alternatively, kit fox populations have declined in past decades and have been the focus of conservation efforts by state and federal agencies [30–34]. While kit foxes were historically trapped and hunted, they were not subjected to intense exploitation and targeted removal as were coyotes. Kit foxes are generally less wary than coyotes [23, 35] and are innately investigative towards novel stimuli [36–40], consistent with the a species that has experienced less intense exploitation.

Anthropogenic pressure may influence behavior of coyotes and kit foxes differently in areas where hunting and trapping occur compared to areas where they are prohibited [9]. We evaluated behavioral differences between coyotes and kit foxes to novel stimuli at 660 scent stations across Utah in areas with and without anthropogenic pressure. We predicted that 1) kit foxes would be more investigative than coyotes in general and 2) coyotes would be less investigative towards novel stimuli in unprotected areas than protected areas, whereas kit foxes would exhibit no difference.

## Methods

### Ethics statement

Fieldwork was approved and sanctioned by United States Department of Defense (DoD) and Utah Division of Wildlife Resources and conducted in compliance with the Institutional Animal Care and Use Committee of Brigham Young University.

### Study areas

We conducted our research at nine study areas across southwestern Utah, U.S.A. (Fig 1), where coyotes and kit foxes are sympatric [41]. Sites were located in the West desert and throughout the southern half of the state. Study areas were in arid landscapes, however, climatic conditions varied between sites. Two study areas were on DoD land where hunting and trapping was prohibited. We considered DoD areas "protected", whereas remaining areas were on public land and allowed hunting and trapping.

### Data collection

To monitor the behavior of canids, we created a grid of sample cells with forced minimum distance of 4 km between cells [42] except on military test ranges (due to safety concerns and site-specific protocols, forced minimum distance was restricted to 1.61 km). A 2.6 km radius buffer was used around each point. We selected this distance based on the square root of home ranges for coyotes occupying similarly semi-arid environments as it reflects daily movement [43–45]. We deployed scent stations between May 2015 and October 2016. To promote independence,

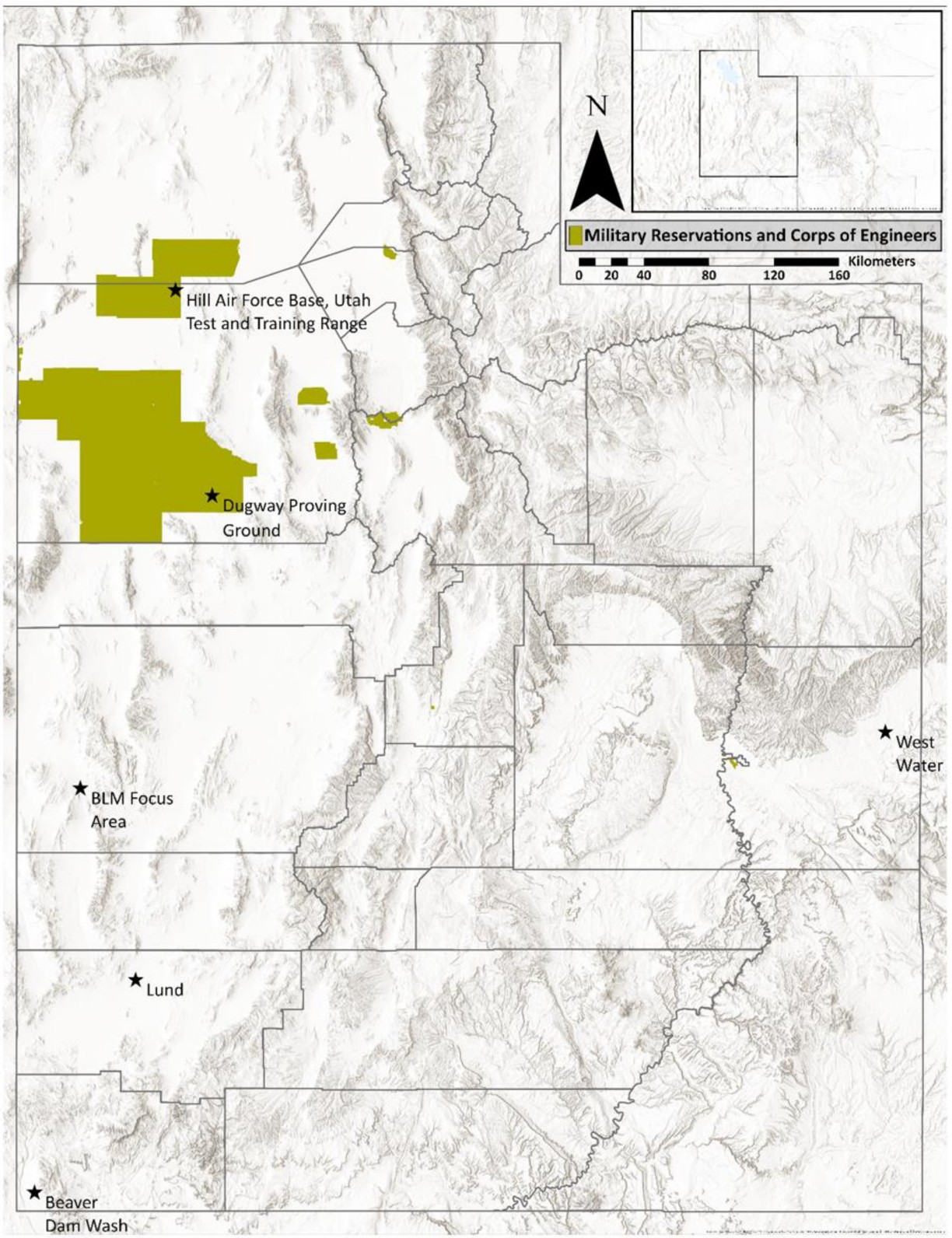

**Fig 1. Map of study area comparing behavioral differences at scent stations between coyotes and kit foxes throughout Utah, USA.** Black stars indicate sampling areas for camera grid. Protected areas are denoted with green shading and used for comparison of behavior between areas with hunting/trapping and without.

we placed cameras within a 300 m buffer of the cell's centroid. Scent stations consisted of an infrared camera (Reconyx© PC900) attached to a post, positioned 27 cm above ground. Cameras were motion-activated and captured images when movement was detected. We randomly assigned every station one of three possible novel objects: pre-scented plaster of Paris tablet with fatty acid lure (Pocatello Supply Depot, Pocatello, Idaho), bundle of nine cotton swabs, or a hollowed golf ball mounted on a wooden dowel. Cotton swabs and golf balls were impregnated with Red and Gray Fox or Willey liquid lure (Murray's Lures, Walker, West Virginia, USA). Attractants were positioned two meters from the camera. Prior research showed no difference in detection between objects or species [46]; additionally, objects were randomly assigned to avoid bias. Scents were refreshed after one week and monitored for an additional week. We recognize the potential influence that vegetation has on the behavior of wildlife, thus, we accounted for differences in vegetation using Landscape Fire and Resource Management Planning Tools (LANDFIRE) data provided by U.S. Forest Service and U.S. Department of the Interior [47]. We classified vegetation as barren (16%), shrub (67%), exotic herbaceous (13%), conifer (2%), or unknown (2%).

To analyze canid behavior, we initially separated photographs by species and classified proximity to stimulus as close (within a one meter) or far. We then classified behavior as investigative or non-investigative. Photographs were considered investigative when behaviors conveyed attention toward the stimulus (scented object or camera; Fig 2). Investigative behaviors included approaching, sniffing or biting the object, or scent marking by urinating or rubbing against the object. Photographs were considered non-investigative when animals displayed no attention to stimuli but remained within the field of view (Fig 2). Repeated visits may have led to increasingly investigative behavior. However, we were unable to identify individuals and subsequent photos showing investigative behavior would have also been included in the analysis. To ensure consistency when categorizing behavior, one technician first categorized photographs as close or far and another technician categorized photographs as investigative or non-investigative. All processing of photographs was conducted by individuals familiar with the study design and trained to identify photographs by proximity or behavior; photographs were randomly selected to validate classifications.

### Statistical analysis

We used mixed-model linear regression to determine the behavior of canids toward novel stimuli. We used proportion of investigative photographs and proportion of close photographs as separate response variables and evaluated the same set of twelve *a priori* models for both responses (Table 1). We accounted for variation across study areas using random effects in the lme4 package [48] in Program R [49]. We evaluated candidate models using conditional Akaike's information criterion (cAIC), which is appropriate for evaluating relative fit among mixed-effects models [50, 51]. To evaluate significance of covariates, we examined overlap in 85% confidence intervals around mean estimates [52].

### Results

Coyotes and kit foxes visited 183 of 660 (~28%) scent stations. We recorded 4,142 photographs of both species and identified 1,008 separate visits. Of the total visits, 217 were of coyotes (73% on protected land, 27% unprotected on land) and 791 were of kit foxes (77% on protected land, 23% on unprotected land).

Our results suggested that canid behaviors differed according to species and land ownership. We found strong support for species and protected areas explaining variation in the proportion of close photographs per visit (3 models with ΔcAIC < 4 included combinations of

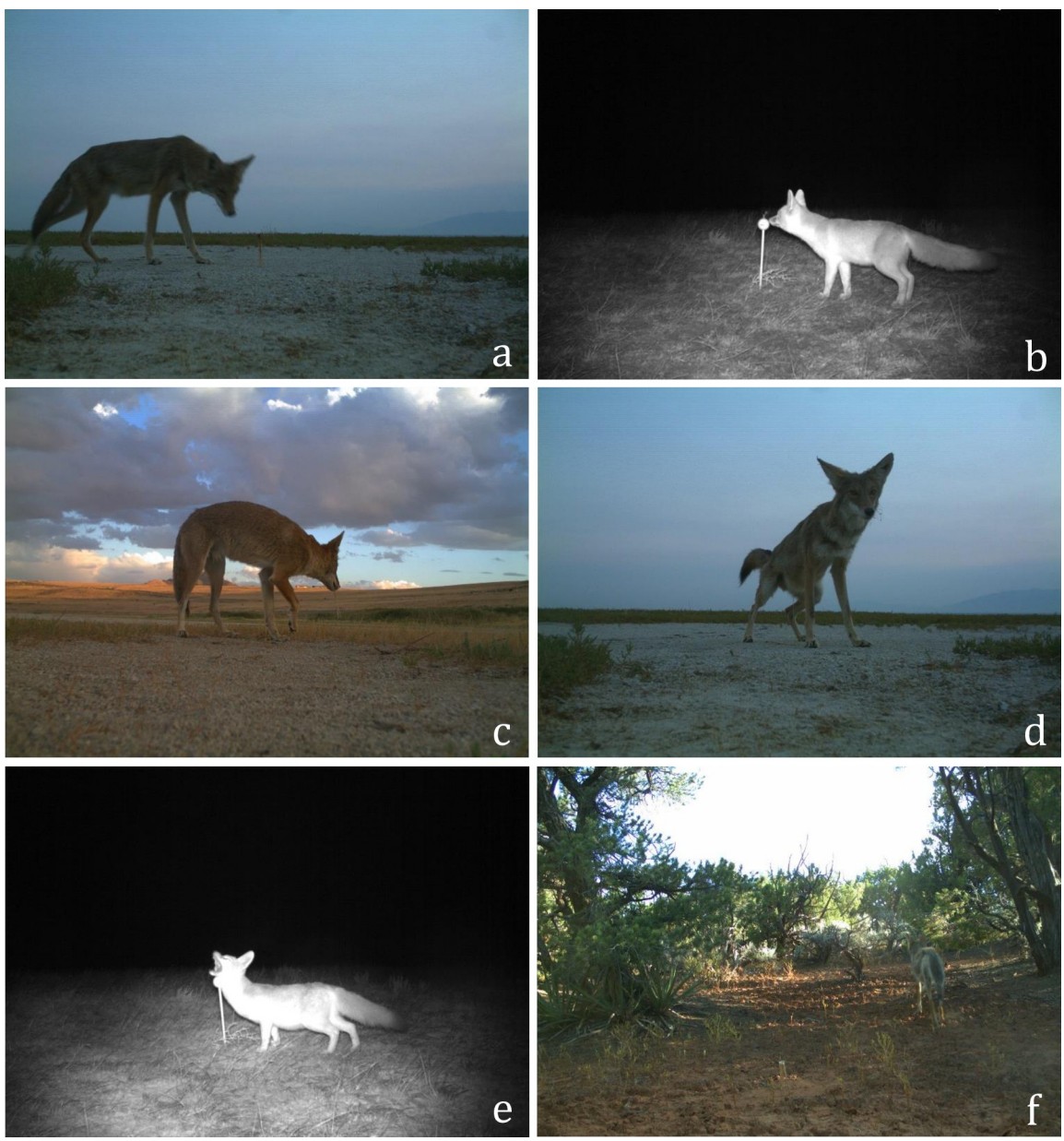

**Fig 2. Ethogram of behaviors investigated in this study.** From top left to bottom right, panel A shows a coyote approaching the lure; in panel B, a kit fox is sniffing the lure; panel C shows a coyote biting the lure; in panel D, a coyote is urinating on the lure; in panel E, a kit fox is rubbing against the object. Panels A-E are examples of the different behaviors categorized as investigative. In panel F, a coyote is near the lure but not interacting with it (classified as non-investigative).

species and protected area fixed effects, combined $w_i$ of these models = 0.90; Table 1A). The most-supported model for proportion of close photographs per visit included additive effects of species and protected areas ($w_i$ = 0.45), and this response (mean ± SE) was greater for kit foxes (0.56 ± 0.02; 85% CI = 0.52–0.60) than for coyote (0.29 ± 0.03; 85% CI = 0.24–0.35; Fig 3), consistent with predictions. Proportion of close photographs per visit was greater on protected areas (0.46 ± 0.03; 85% CI = 0.40–0.51) than on unprotected areas (0.40 ± 0.02; 85% CI = 0.36–0.43) for both species, though 85% confidence intervals overlapped.

**Table 1. Model selection comparing proportion of close photographs (A) and investigative photographs (B) of kit foxes and coyotes at scent stations throughout Utah, USA.** Top models suggested differences between species and between protected versus unprotected land. Table contains conditional Akaike Information Criteria (cAIC) and Akaike model weight ($w_i$) and conditional log-likelihood (LL) of candidate models.

| | Model | LL | K | cAIC | ΔcAIC | $w_i$ |
|---|---|---|---|---|---|---|
| (A) | Species + Protected | -423.10 | 5 | 954.35 | 0.00 | 0.45 |
| | Species | -424.91 | 4 | 955.64 | 1.29 | 0.24 |
| | Species * Protected | -422.61 | 6 | 955.94 | 1.59 | 0.20 |
| | Species + Protected + Vegetation | -421.32 | 9 | 959.17 | 4.82 | 0.04 |
| | Species + Vegetation | -422.89 | 8 | 959.72 | 5.37 | 0.03 |
| | Species * Protected + Vegetation | -420.89 | 10 | 960.79 | 6.44 | 0.02 |
| | Species * Vegetation + Protected | -418.44 | 13 | 962.10 | 7.75 | 0.01 |
| | Species * Vegetation | -420.10 | 12 | 963.25 | 8.90 | 0.01 |
| | Protected | -428.46 | 4 | 1000.17 | 45.82 | 0.00 |
| | Intercept | -429.64 | 3 | 1000.28 | 45.93 | 0.00 |
| | Vegetation | -427.15 | 7 | 1002.48 | 48.13 | 0.00 |
| | Vegetation + Protected | -425.90 | 8 | 1003.11 | 48.76 | 0.00 |
| (B) | Species + Protected | -407.32 | 5 | 873.63 | 0.00 | 0.52 |
| | Species * Protected | -406.96 | 6 | 875.45 | 1.82 | 0.21 |
| | Protected | -410.93 | 4 | 877.12 | 3.49 | 0.09 |
| | Species | -406.32 | 4 | 877.25 | 3.62 | 0.09 |
| | Species + Protected + Vegetation | -404.72 | 9 | 879.09 | 5.46 | 0.03 |
| | Intercept | -409.74 | 3 | 880.23 | 6.60 | 0.02 |
| | Species * Protected + Vegwgetation | -404.42 | 10 | 880.93 | 7.30 | 0.01 |
| | Species + Vegetation | -405.82 | 8 | 881.45 | 7.82 | 0.01 |
| | Vegetation + Protected | -408.71 | 8 | 883.25 | 9.62 | 0.00 |
| | Vegetation | -409.76 | 7 | 884.71 | 11.08 | 0.00 |
| | Species * Vegetation + Protected | -401.66 | 13 | 885.08 | 11.45 | 0.00 |
| | Species * Vegetation | -402.76 | 12 | 887.69 | 14.06 | 0.00 |

Similarly, we found support for differences in proportion of investigative photographs per visits between protected and unprotected areas and species (4 models with ΔcAIC < 4 included combinations of species and protected area fixed effects, combined $w_i$ = 0.91; Table 1B). The most-supported model for proportion of investigative photographs per visit again included additive effects of species and protected areas ($w_i$ = 0.52). Kit foxes were more investigative than coyotes (difference in means = 0.07, 85% CI = 0.03–0.12), and both species were more investigative on protected lands than unprotected lands (difference in means = 0.08, 85% CI = 0.03–0.12).

## Discussion

We observed behavioral differences between coyotes and kit foxes suggesting coyotes were the more wary species, consistent with our predictions. Coyotes maintained a greater distance from novel stimuli and interacted with stimuli (e.g., biting, urinating, defecating on scent) less often than kit foxes. Coyotes increased averseness towards novel stimuli in unprotected areas, supporting our predictions. While anthropogenic activity still occurs on protected land, levels of recreation did not substantially influence surrounding wildlife [53]. Our results are consistent with previous research describing heightened aversiveness of coyotes to anthropogenic pressure [9].

Kit foxes were more also investigative in protected areas than unprotected, suggesting that increased anthropogenic pressure may result in increased neophobia. Overall, kit foxes were

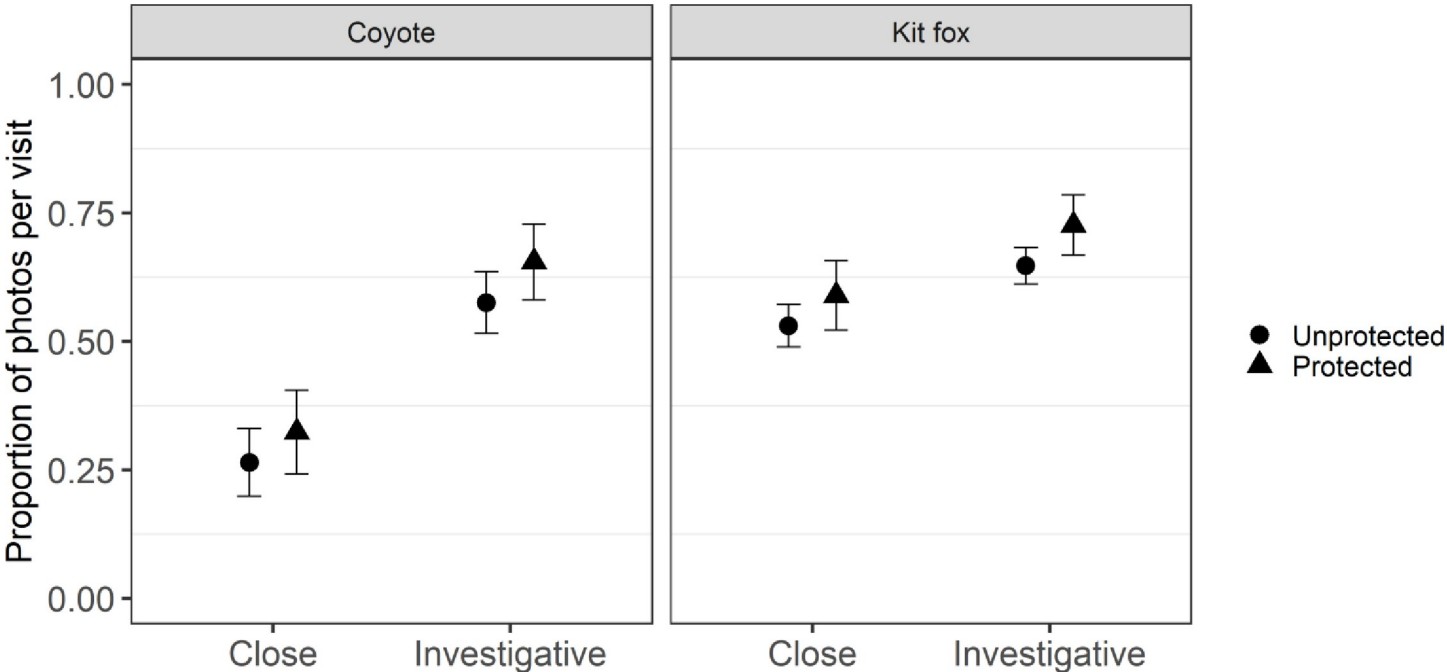

**Fig 3. Proportion of investigative and close (within one meter) photographs per visit (± 85% CI) by kit fox (*Vulpes macrotis*) and coyote (*Canis latrans*) at scent stations in Utah (2015–2016).** Protected sites were on Department of Defense land, where hunting and trapping were not allowed. Unprotected sites were on public land and permitted harvest activities.

more investigate and maintained a closer distance to stimuli than coyotes. Contrasting histories of anthropogenic pressure may cause differences in behavior between species. However, the observed differences between species may be caused by other factors, as well. Underlying differences in social structure and landscape use may also affect behavior [25, 54]. Additionally, coyotes often represent the leading source of mortality for kit foxes [13, 55] and as such, coyote activity can influence habitat use and detection probability of kit foxes [12]. Additionally, individual personality and past experiences with anthropogenic stimuli may impact behavioral responses [4].

Numerous factors may influence the exploratory behavior of canids. Differences in behavior between individuals may have been related to social status or trophic level. Socially dominant coyotes were less neophobic in captivity; however, these characteristics may be selected against in the wild through predator control [1]. Higher trophic levels were associated with decreased neophobia [2]; however, we found that coyotes interacted with novel objects less than kit fox. Familiarity of the areas and levels of disturbance may also have influenced exploratory behavior. Coyotes in unfamiliar areas showed decreased neophobia compared to areas within their home range [6]. Spotted hyenas (*Crocuta crocuta*) exhibited less neophobia in areas with high levels of anthropogenic disturbance [5]. Developmental differences between species may also influence interactions with novel stimuli. Differences in motor skills and developmental trajectories led to wolves (*C. lupus*) interacting with novel objects and environments more than dogs [C. l. familiaris; 56]; however, habituation has led to decreased neophobia in dogs but not wolves [57]. Size of the object and duration of exposure can also influence the extent of exploratory behavior by canids. Coyotes interacted with smaller novel objects more than large objects, however, this effect reversed after objects were removed [24]. Similarly, Culpeo fox (*Lycalopex culpaeus*) and grey foxes (*Urocyon cinereoargenteus*) increased

exploration after novel objects were removed, despite initial neophobic responses from culpeo foxes [58]. Prior studies have highlighted the complexity of factors governing behavioral responses of canids to novel objects.

Anthropogenic pressures can affect various behaviors including mating, survival, social structure, and foraging of wildlife [7], often leading to increased wariness of anthropogenic stimuli [9, 59]. We provide additional research on the behavior of coyotes and kit foxes, highlighting behavioral differences between species in areas with and without hunting/trapping. Both species were more investigative on protected land than unprotected land. Coyotes maintained a greater distance from novel objects and were generally less investigative than kit foxes, potentially due to extensive exploitation causing a general increase in wariness of anthropogenic objects [4]. Our findings provide a behavioral basis for the commonly held notion that coyotes are more difficult to trap. Kit foxes were more investigative than coyotes, particularly on protected land, suggesting a greater sensitivity to anthropogenic pressure than coyotes. As kit foxes are a species of conservation concern, these results may be relevant to management efforts in areas of high disturbance. Our findings provide additional evidence that anthropogenic pressure can alter the fine-scale behavior of wildlife species.

## Acknowledgments

We would like to acknowledge the Bureau of Land Management, Utah Department of Natural Resources, US Army Dugway Proving Ground, US Hill Air Force Base, Brigham Young University, and all of our technicians for their support and assistance on this project.

## Author Contributions

**Conceptualization:** Kelsey A. Richards, Lucas K. Hall, Randy T. Larsen.

**Data curation:** Maksim Sergeyev, Kelsey A. Richards, Kristen S. Ellis, Lucas K. Hall, Jason A. Wood.

**Formal analysis:** Kristen S. Ellis.

**Investigation:** Kelsey A. Richards.

**Methodology:** Kelsey A. Richards, Kristen S. Ellis, Lucas K. Hall, Randy T. Larsen.

**Project administration:** Randy T. Larsen.

**Resources:** Randy T. Larsen.

**Supervision:** Randy T. Larsen.

**Writing – original draft:** Maksim Sergeyev, Lucas K. Hall, Jason A. Wood.

**Writing – review & editing:** Maksim Sergeyev, Lucas K. Hall, Randy T. Larsen.

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
