## [Decision Letter · Decision Letter 0]

11 Mar 2020

PONE-D-19-32511

Behavioral differences at scent stations between two exploited species of desert canids

PLOS ONE

Dear Maksim Sergeyev,

Thank you for submitting your manuscript to PLOS ONE. After careful consideration, we feel that it has merit but does not fully meet PLOS ONE’s publication criteria as it currently stands. Therefore, we invite you to submit a revised version of the manuscript that addresses the points raised during the review process.

I suggest the authors pay much attention to the reviewer's comments and suggestion.

We would appreciate receiving your revised manuscript by Apr 25 2020 11:59PM. To enhance the reproducibility of your results, we recommend that if applicable you deposit your laboratory protocols in protocols.io, where a protocol can be assigned its own identifier (DOI) such that it can be cited independently in the future. For instructions see: http://journals.plos.org/plosone/s/submission-guidelines#loc-laboratory-protocols

We look forward to receiving your revised manuscript.

Kind regards,

De-Hua Wang, Ph.D.

Academic Editor

PLOS ONE

Journal Requirements:

2) In your Methods section, please provide additional location information of the study areas, including geographic coordinates for the data set if available.

3) Please include your tables as part of your main manuscript and remove the individual files. Please note that supplementary tables (should remain/ be uploaded) as separate "supporting information" files

4)  We note that you have indicated that data from this study are available upon request. PLOS only allows data to be available upon request if there are legal or ethical restrictions on sharing data publicly. For more information on unacceptable data access restrictions, please see http://journals.plos.org/plosone/s/data-availability#loc-unacceptable-data-access-restrictions.

5) Thank you for stating the following in the Acknowledgments Section of your manuscript:

[We are grateful to Bureau of Land Management, Utah Department of Natural Resources,

US Army Dugway Proving Ground, US Hill Air Force Base, and Brigham Young University for

funding this research and thank our technicians for their assistance.]

 [The author(s) received no specific funding for this work.]

Please include the updated Funding Statement in your cover letter. We will change the online submission form on your behalf.

Reviewers' comments:

Reviewer's Responses to Questions

**Comments to the Author**

1. Is the manuscript technically sound, and do the data support the conclusions?

Reviewer #1: Yes

2. Has the statistical analysis been performed appropriately and rigorously? 

Reviewer #1: Yes

3. Have the authors made all data underlying the findings in their manuscript fully available?

Reviewer #1: Yes

4. Is the manuscript presented in an intelligible fashion and written in standard English?

Reviewer #1: Yes

5. Review Comments to the Author

Reviewer #1: This is a very interesting study in which the authors tried to explore the different behavioral responses of coyotes and kit foxes to novel objects within and outside the protected area by infra-cameras. The idea of this study is innovative and sharp. The manuscript is short, simple, and well written. The results are promising. I have no further comments and questions about its acceptance except the following questions:

1) It is good to compare the two species’ behavioral responses to novel objects within and outside the protected area, and you have provided a criteria of 2.6km radius buffer based on the home ranges of the two species. However, I am thinking the dispersal distance of each species might be more important than that of the home range/daily movement, due to the fact that you are studying the long-term effects/pressures of anthropogenic activities. Thus, the grouping of putting the site “Thomas Range” into unprotected area (if my understanding is right) might be a problem. I would suggest you re-analysis your data by excluding this part of data from your whole data set to see if you still get the same result. Then, you can soundly make the statement.

2) Additional information about the two technicians may be needed for categorizing photographs as close/far and investigative or non-investigative. Did they clearly know the experimental design and grouping of the scents in each photograph? How did you manage to reduce the observer’s bias in doing so?

3) You mentioned in the results section (Line 5 on page 8)” We identified trends suggesting coyotes were warier on protected lands than unprotected land, however these trends were not statistically significant and were consistent for both species”. What does this mean or what are the possible causes for this result? It might be better for readers to understand to provide some explanations in the discussion section.

6. PLOS authors have the option to publish the peer review history of their article (what does this mean?). If published, this will include your full peer review and any attached files.

Reviewer #1: No

---

## [Author Response · Author response to Decision Letter 0]

25 Mar 2020

Thank you to the editor and reviewer for providing comments on our manuscript. We have incorporated the suggested changes and have detailed the edits made in the 'Response to Reviewers' document.

---

## [Editor Report · Decision Letter 1]

16 Apr 2020

Behavioral differences at scent stations between two exploited species of desert canids

PONE-D-19-32511R1

Dear Dr. Maksim Sergeyev,

We are pleased to inform you that your manuscript has been judged scientifically suitable for publication and will be formally accepted for publication once it complies with all outstanding technical requirements.

With kind regards,

De-Hua Wang, Ph.D.

Academic Editor

PLOS ONE

Additional Editor Comments (optional):

Authors have revised their ms according to the comments and suggestion from the reviewers. I have no further comments for this ms. I recommend this ms can be accepted for pulication in Plos One.
---

## [Editor Report · Acceptance letter]

22 Apr 2020

PONE-D-19-32511R1 

Behavioral differences at scent stations between two exploited species of desert canids 

Dear Dr. Sergeyev:

I am pleased to inform you that your manuscript has been deemed suitable for publication in PLOS ONE. Congratulations! Your manuscript is now with our production department. 

With kind regards,

on behalf of

Prof. De-Hua Wang 

Academic Editor

PLOS ONE